# Numerous inequalities and related communications accompanying discrete divergence models in probability spaces

Vikramjeet Singh[1], Sunil Kumar Sharma[2]*, Om Parkash[3], Mona Bin-Asfour[4]

1 Department of Mathematics, I. K. Gujral Punjab Technical University, Amritsar Campus, Amritsar, Punjab, India, 2 Department of Information Systems, College of Computer and Information Sciences, Majmaah University, Majmaah, Saudi Arabia, 3 Department of Mathematics, Graphic Era Deemed to be University, Dehradun, Uttarakhand, India, 4 Deparment of Mathematics and Statistics, College of Science, Imam Mohammad Ibn Saud Islamic University, Riyadh, Saudi Arabia

* s.sharma@mu.edu.sa

## Abstract

The analysis of inequalities aids in the formulation of optimal coding schemes that improve the rate of information transfer while reducing the probability of errors. This consequently implies that we have a major impact on having solid and secured correspondence systems for applications running from broadcast communications to information pressure. Information theory inequalities constitute a theoretical tool-box for designing channels that can overcome a challenging environment, allowing information to be held and communicated reliably and securely in a world that is becoming ever more interconnected. The paper introduces a general divergence model in general probability spaces and extends known information-theoretic inequalities and results based on variational models. We have built various inequalities for finite sequences of positive real numbers, the specific cases of which are important in information theory, especially in connection with several divergence models that remain in the literature. Additionally, we have derived certain other important communications concerning positive real numbers in relation to some divergence models.

## Introduction

The well-known validity of coding theory allows for the examination of code combinations using discrete probabilistic entropic models and supports applications in many domains. Shannon [1] developed the concept of entropy within discrete probability spaces, forming the foundation of information theory. Shannon's widely accepted view of probabilistic entropy [1] expanded the coding theory literature by introducing several entropic models. This established process laid the groundwork for a discrete entropic model with unexpectedly favourable features.

**Data availability statement:** All relevant data are within the paper.

**Funding:** The author extends the appreciation to the Deanship of Postgraduate Studies and Scientific Research at Majmaah University for funding this research work through the (Project Number R-2026-39).

**Competing interests:** The authors have declared that no competing interests exist. This does not alter our adherence to PLOS ONE policies on sharing data and materials.

We have the notion that $\Upsilon_n = \left\{ (q_1, q_2, ..., q_n) : q_i \geq 0; i = 1,...,n; \sum_{i=1}^{n} q_i = 1 \right\}$ is the assemblage of all unconnected possibility distributions with nonnegative members and complete support on a set of cardinality n and $\Upsilon = \bigcup_{n=1}^{\infty} \Upsilon_n$. A non-degenerate possibility distribution $q_i \in \Upsilon_n$ is thought to be a probability distribution. In many cases, transactions must be delivered using discrete probability distributions, each of which contains a positive real value. Consequently, we define the following sets:

$$\Upsilon_n^* = \left\{ (q_1, q_2, ..., q_n) : q_i > 0; i = 1,...,n; \sum_{i=1}^{n} q_i = 1 \right\}.$$

For every probability distribution $q_i \in \Upsilon_n$, we give various current entropic models:
The Shannon [1] entropy:

$$H(P) = -\sum_{i=1}^{n} p_i \log_2 p_i$$

$$(1.1)$$

The Renyi [2] entropy:

$$H_\alpha(P) = (1-\alpha)^{-1} \log_2 \left( \sum_{i=1}^{n} p_i^\alpha \right), \ \alpha > 0, \alpha \neq 1$$

$$(1.2)$$

The Havrda-Charvat [3] entropy:

$$H^\alpha(P) = (1-2^{1-\alpha})^{-1} \left( 1 - \sum_{i=1}^{n} p_i^\alpha \right), \ \alpha > 0, \alpha \neq 1$$

$$(1.3)$$

Huang and Zhang [4] provided a new interpretation of Shannon's [1] mutual information, noting that it is often difficult to compute for uncountable sample spaces. As a result, the authors observed that the asymptotic modus operandi pedestal on Fisher information often provides spectacular estimates to such information, and they mirrored the estimates pedestal on particular divergence models in the probability spaces. Furthermore, the authors performed numerical repetition and demonstrated that their predicted mode of operation was incredibly wonderful, with increasing ease of application to a wide range of real and hypothetical issues. Furthermore, discrete entropy models have a wide range of applications. Sholehkerdar et al. [5] said that entropy-based measures are often used to evaluate objective picture fusion supremacy due to single layer entropy execution and an inconsequential parameter set. In their communication, the authors made available a hypothetical research of picture fusion quality measures based on Tsallis [6] entropy. The goal of this research was to evaluate if the chosen quality measure was capable of providing the necessary performances predicted from a perfect information-based image fusion eminence metric. To assess the Tsallis [6] excellence metric, the study

results suggested the use of an image realisation model to generate a closed-form appearance for excellence, with weighted averaging used as a fusion process. Their results show that the Tsallis-based excellence measure deviates from expected performance in terms of response to signal-to-noise ratio discrepancy and the effect of entropy order on the measured excellence indicator. Furthermore, the writers conducted evaluations on genuine photographs, the results of which confirmed conformity with the hypothetical exploration. Lenormand et al. [7] discussed the applications of entropy-based models in the urban environment, noting that describing and quantifying spatial inequalities across the urban background remains a complex and mysterious task that has been aided by the increasing availability of large geolocated catalogues. The outcomes of their investigation showed that the entropy-based measure of spatial distribution is a useful indicator of a region's socioeconomic status and related factors. Lu et al. [8] said that measures of expectedness in physiological indicators based on entropy metrics have been widely used in the presenting domains of medical valuation and clinical identification. The authors proposed a novel entropy-based pattern knowledge for valuing physiological indicators that combines single spectrum analysis and entropy metrics. Saraiva, P. [9] provided brief and spontaneous summary of Shannon's [1] entropy, including chosen characteristics, and offered applications of the model in two contrasting viewpoints from its inception: biological variety and a unique research on student mobility. Zhang and Shi [10] emphasized that Shannon's [1] entropy is a fundamental building block of information theory and an essential component of Machine Learning (ML) approaches. The authors created asymptotic holdings that do not need any conventions on the original propagation, and these features allow for interval estimation and statistical tests with full Shannon entropy. Elgawad et al. [11] applied Shannon's [1] entropy to statistics, including order statistics and several known distributions. Stoyanov et al. [12] employed maximum entropy technique to develop a unique concept for the M-indeterminacy of disseminations on the positive half-line (Stieltjes case) and then derived some important conclusions. Furthermore, the authors proved how the maximum entropy is connected to the symmetry characteristics and M-indeterminacy Wang et al. [13] and others are among the pioneers who have publicly stated their commitment to studying entropic models. In the following, we discuss the essential perspective of the discrete inaccuracy model as attributed to Kerridge [14].

Assume that an experimenter says that the chance of the result of a random experiment is, despite the true probability being. Then, utilizing certain compelling postulates, Kerridge [14] proved that the inaccuracy of the aforementioned claim is dictated by the later numerical exterior:

$$I(\mathrm{P};\mathrm{Q}) = -\sum_{i=1}^{n} p_i \log_2 q_i$$

(1.4)

Sathar et al. [15] explored the prior inaccuracy model and therefore presented nonparametric estimators for these models. The authors carefully explored the asymptotic behaviour of these estimators under varied appropriateness and reliability requirements. Additionally, the authors examined the performance of the projected estimators using the Monte-Carlo simulation technique. Nath [16] also suggested the non-additive inaccuracy model predetermined by the following manifestation:

$$I^{\alpha}(\mathrm{P};\mathrm{Q}) = (1 - 2^{1-\alpha})^{-1} \left( 1 - \sum_{i=1}^{n} p_i q_i^{\alpha-1} \right)$$

(1.5)

where $\alpha > 0$, $\alpha \neq 1$, $n \geq 1$ an integer.

Different instigators proposed novel inaccuracy models because of their application in statistics, coding theory, and other related subjects. Kapur [17], Molloy and Ford [18], and others have made contributions to the characterisation and implementation of inaccuracy models.

The appreciation of entropic models has been investigated as measures of the amount of information for a given probability distribution; it is customary to scrutinize such models in order to assess the amount of information shared between two probability distributions in the sense of how close the two distributions are to one another. Such measurements are known as distance models, and the perception of distance is one of the most essential and fundamental ideas in the various applications of information theory, which has substantial implications in a wide variety of mathematical sciences domains.

Another opinion on the importance of such metrics is that several efforts have been made to broaden the idea of distance in domains other than mathematics. Disciplines such as economics, sociology, psychology, linguistics, genetics, and biology may all benefit from distance measurement. However, distance in such circumstances is not always geometrical, hence there is a need for modification when it comes to distance in probability spaces. Here, we emphasize the need of making adjustments while considering the distance model in probability space. To overcome these limitations, several novel divergence metrics have been proposed and defined. The most important and practically constructive divergence model is attributed to Kullback and Leibler [19], as defined by the following formula.:

$$D(P:Q) = \sum_{i=1}^{n} p_i \ln \frac{p_i}{q_i}$$

(1.6)

The following generalized parametric models of directed divergence and equipped them with applications in the disciplines of statistics and operations research.

$$S(P:Q) = \sum_{i=1}^{n} \left[ \left( \frac{1}{\sqrt{p_i}} + \frac{1}{\sqrt{q_i}} \right) \left( \frac{p_i + q_i}{2} \right)^{3/2} - 2p_i \right]$$

(1.7)

$$G^{\alpha}(P:Q) = \frac{\sum_{i=1}^{n} p_i \left( \alpha + \frac{1}{2} \right)^{\log \frac{p_i}{q_i}} - 1}{\alpha - \frac{1}{2}}, \alpha > \frac{1}{2}$$

(1.8)

where $\alpha$ is a real parameter.

$$D_{\alpha,\beta}(P:Q) = \frac{1}{\alpha - \beta} \left( \sum_{i=1}^{n} p_i^{\alpha-\beta+1} q_i^{\beta-\alpha} - 1 \right), \alpha \neq \beta, \beta < \alpha + 1, \alpha > 0, \beta > 0$$

(1.9)

Parkash and Kakkar [20] developed the following extended parametric models of directed divergence and used them in the area of coding theory.

$$_{\alpha}D(P:Q) = \frac{\sum_{i=1}^{n} p_i \alpha^{\ln \frac{p_i}{q_i}} - 1}{\alpha - 1}, \quad \alpha > 1$$

(1.10)

$$D_{\alpha}(P:Q) = \frac{1}{\alpha - 1} \left( \prod_{i=1}^{n} \left( \frac{p_i}{q_i} \right)^{-p_i(1-\alpha)} - 1 \right), \quad \alpha > 1$$

(1.11)

$$_\alpha A(P:Q) = \frac{1}{\alpha-1}\sum_{i=1}^{n} p_i \alpha^{\ln p_i}\left(\alpha^{\ln\frac{1}{q_i}}-1\right), \ \alpha > 1$$

(1.12)

Pronzato et al. [21], Kumari and Sharma [22], Torra et al. [23], Khalaj et al. [24], Dwivedi et al. [25], Nielsen [26], Fukumizu [27], Li et al. [28], and others have all contributed to the suggestions, characterizations, generalizations, and implementations of divergence models. In the sequel, we illustrate some novel implications of divergence models. To satisfy our aim, we need the following standard findings from Ash [29] and Cover and Thomas [30]:

### Research gap and Motivation

Although many divergence measures — especially the Kullback-Leibler and Rényi divergences — have received a great deal of attention, the current work tends to focus along the lines of properties or special cases of the divergences rather than an unified perspective on their relations through inequalities. The lack of cohesive inequalities linking these variables constrains their comparative analysis and practical interpretation across various probability models. Driven by this need, the current study formulates novel inequalities that generalize and expand existing divergence functions, hence providing enhanced understanding of the architecture of discrete probability spaces. This paper delineates new correlations among generalized divergence measures, offers a formal demonstration of the monotonic behaviour of Rényi divergence via inequality (2.7), and presents a symmetric divergence measure $J_\alpha \ (P;Q)$ with enhanced analytical features. These findings jointly augment the theoretical foundation of divergence-based modelling and bolster its prospective applications in communication theory, coding systems, and statistical inference.

The impetus for creating new inequalities among these divergence metrics stems from their capacity to provide more precise limits and enhanced performance estimates for practical applications, including source coding, sensor data fusion, and statistical inference. Establishing these links connects abstract mathematical analysis to measurable improvements in practical systems.

Unless otherwise stated, throughout the next portions of the article, We will assume that $X = (x_1, x_2, ..., x_n)$, $Y = (y_1, y_2, ..., y_n)$, $A = (a_1, a_2, ..., a_n)$ and $B = (b_1, b_2..., b_n)$ and are positive real values.

**Theorem 1.1**. With the above professed conventions, the subsequent inequality grips:

$$\left(\sum_{i=1}^{n} x_i\right)\log_2\left(\frac{\sum_{i=1}^{n} x_i}{\sum_{i=1}^{n} y_i}\right) \leq \sum_{i=1}^{n} x_i\log_2\left(\frac{x_i}{y_i}\right)$$

(1.13)

for all integers $n \geq 1$. If $n = 1$, then (1.13) holds only as an equality.

**Theorem 1.2.** With the above professed conventions and for $n \geq 1$ if $\sum_{i=1}^{n} a_i \geq \sum_{i=1}^{n} b_i$, then we attain the successive in equation:

$$\sum_{i=1}^{n} a_i\log_2\left(\frac{1}{a_i}\right) \leq \sum_{i=1}^{n} a_i\log_2\left(\frac{1}{b_i}\right)$$

(1.14)

with the sign of equality iff $a_i = b_i$.

Theorem 1.3. For $P \in \Gamma_n^*, Q \in \Gamma_n^*$, the subsequent inequality holds good:

$$\frac{1}{\alpha-1}\log_2\left(\sum_{i=1}^{n} p_i^\alpha q_i^{1-\alpha}\right) > 0$$

(1.15)

for $\alpha > 0$, $\alpha \neq 1$ and (1.15) holds except when $p_i = q_i$ $\forall i$.

Theorem 1.4. With the above professed conventions and for $n \geq 2$, the succeeding inequalities grip:

$$\left(\sum_{i=1}^{n} x_i\right)^{\alpha} \left(\sum_{i=1}^{n} y_i\right)^{1-\alpha} < \sum_{i=1}^{n} x_i^{\alpha} y_i^{1-\alpha} \text{ if } \alpha > 1 \tag{1.16}$$

and

$$\left(\sum_{i=1}^{n} x_i\right)^{\alpha} \left(\sum_{i=1}^{n} y_i\right)^{1-\alpha} > \sum_{i=1}^{n} x_i^{\alpha} y_i^{1-\alpha} \text{ if } 0 < \alpha < 1 \tag{1.17}$$

except when $\frac{x_1}{y_1} = \frac{x_2}{y_2} = ... = \frac{x_n}{y_n}$.

Theorem 1.5. With the above professed conventions and for $n \geq 1$ a given integer, the subsequent inequality clutches good:

$$\frac{\sum_{i=1}^{n} x_i \log_2 y_i}{\sum_{i=1}^{n} x_i} \leq \log_2 \left(\frac{\sum_{i=1}^{n} x_i y_i}{\sum_{i=1}^{n} x_i}\right) \tag{1.18}$$

If $n = 1$, then (1.18) clutches as an equivalence. If $n \geq 2$, then the insignia of equivalence in (1.18) grips only for the equivalence in $y_i$.

### Various inequalities and other communications associated with divergence models

In this subdivision, we derive some inequalities and other communications concerning positive real numbers and deliberate their significance in relation to some divergence models.

Let $P \in \Gamma_n^*$, $Q \in \Gamma_n^*$, $n \geq 1$ an integer.

**Definition 2.1.** A function $\varphi : \Gamma_n^* \times \Gamma_n^* \to R$, $n \geq 1$ an integer, is supposed to be a divergence function if, for all $P \in \Gamma_n^*$, $Q \in \Gamma_n^*$,

$$\varphi(P; Q) \geq 0 \tag{2.1}$$

and

$$\varphi(P; Q) = 0 \text{ iff } p_i = q_i \tag{2.2}$$

If, in addition, we have the subsequent manifestation:

$$\varphi(P; Q) = \varphi(Q; P) \tag{2.3}$$

then $\varphi$ is said to be a symmetric divergence function.

**Example 2.2.** Choose $\varphi = D$

where $D : \Gamma_n^* \times \Gamma_n^* \to R$, $n \geq 1$ an integer; is demarcated by the subsequent expression:

$$D(\mathrm{P}; \mathrm{Q}) = \sum_{i=1}^{n} p_i \log_2 \left( \frac{p_i}{q_i} \right).$$

(2.4)

This divergence function is acknowledged as Kullback-Liebler's [19] divergence function.

Choosing $x_i = p_i$, $y_i = q_i$, $P \in \Gamma_n^*$, $Q \in \Gamma_n^*$, $n \geq 1$ an integer. If $n = 1$, then $D(1; 1) = 0$. If $n \geq 2$, then by using Theorem 1.1, it follows that $\sum_{i=1}^{n} p_i \log_2 \frac{p_i}{q_i} \geq 0$ with the insignia of equivalence iff $p_i = q_i$.

Consequently, $D(\mathrm{P}; \mathrm{Q}) \geq 0$

For $n \geq 2$, it has numerous interesting interpretations: Firstly, it is acknowledged as the distance of one probability distribution $P \in \Gamma_n^*$ from another probability distribution $Q \in \Gamma_n$. Secondly, it is interpreted as a measure of inaccuracy of error due to Kerridge [14]. Thirdly, it is interpreted as a measure of inefficiency due to Cover and Thomas [30] of assuming the probability distribution $Q \in \Gamma_n^*$ when, in representativeness, the true probability distribution is $P \in \Gamma_n^*$. Fourthly, it is likewise entitled the Shannon's [1] information gain when the probability distribution $Q \in \Gamma_n^*$ is replaced by the probability distribution $P \in \Gamma_n^*$.

**Definition 2.3.** With the above professed conventions and for $n \geq 1$ an integer, if $\sum_{i=1}^{n} x_i \geq \sum_{j=1}^{n} y_j$, then the directed divergence denoted by $\overline{D}(\mathrm{X}; \mathrm{Y})$ of the sequence $X$ from the sequence $Y$ is demarcated by the succeeding manifestation:

$$\overline{D}(\mathrm{X}; \mathrm{Y}) = \frac{\displaystyle\sum_{i=1}^{n} x_i \log_2 \left( \frac{x_i}{y_i} \right)}{\displaystyle\sum_{i=1}^{n} x_i}.$$

(2.5)

Making practice of Theorem 1.2, it shadows that

$\overline{D}(\mathrm{X}; \mathrm{Y}) \geq 0$ with the emblem of equivalence iff $x_i = y_i$.

Accordingly $\overline{D}(\mathrm{X}; \mathrm{Y})$ appears to be an appropriate generalization of $D(\mathrm{P}; \mathrm{Q})$ when $P \in \Gamma_n^*$, $Q \in \Gamma_n^*$.

Now, Renyi 's [2] divergence models is specified by the succeeding expression:

$$D_\alpha(\mathrm{P}; \mathrm{Q}) = (\alpha - 1)^{-1} \log_2 \left( \sum_{i=1}^{n} p_i^\alpha q_i^{1-\alpha} \right); \alpha > 0, \alpha \neq 1$$

(2.6)

The measure $D_\alpha(\mathrm{P}; \mathrm{Q})$ is entitled as the information gain of order $\alpha$, $\alpha > 0$, $\alpha \neq 1$, when the probability distribution $Q \in \Gamma_n^*$ is replaced by the probability distribution $P \in \Gamma_n^*$, $n \geq 2$ an integer. We observe that

$\lim_{\alpha \to 1} D_\alpha(\mathrm{P}; \mathrm{Q}) = D(\mathrm{P}; \mathrm{Q})$, which is an expression termed as Shannon's [1] information gain.

Consequently, $D(\mathrm{P}; \mathrm{Q})$ may possibly be regarded as the information gain of order 1 and accordingly, it may be written as $D_1(\mathrm{P}; \mathrm{Q})$ depending upon the situation.

Renyi [2] investigated and ascertained that $D_\alpha(\mathrm{P}; \mathrm{Q})$ with the emblem of equivalence iff $p_i = q_i$, $n \geq 2$ an integer. We contribute with an alternative proof of this consequence. Keeping in understanding equation (1.15), it is sufficient to confirm that $D_\alpha(\mathrm{P}; \mathrm{Q}) = 0$ iff $p_i = q_i$.

If $p_i = q_i$, $i = 1, 2, ..., n$; then $D_\alpha(\mathrm{P}; \mathrm{Q}) = 0$.

Now suppose that $D_\alpha(\mathrm{P}; \mathrm{Q}) = 0$. Then $-\log_2 \sum_{i=1}^{n} p_i^\alpha q_i^{1-\alpha} = 0$.

Also, by Theorem 1.1, $\sum_{i=1}^{n} p_i \log_2 \left( \frac{p_i}{p_i^{\alpha} q_i^{1-\alpha}} \right) = -\log_2 \sum_{i=1}^{n} p_i^{\alpha} q_i^{1-\alpha}$ iff $\frac{p_1}{p_1^{\alpha} q_1^{1-\alpha}} = \ldots = \frac{p_n}{p_n^{\alpha} q_n^{1-\alpha}}$, that is, $p_i = q_i$, as $\alpha > 0, \alpha \neq 1$.

Since $-\log_2 \left( \sum_{i=1}^{n} p_i^{\alpha} q_i^{1-\alpha} \right) = 0$ and $\sum_{i=1}^{n} p_i \log_2 \left( \frac{p_i}{p_i^{\alpha} q_i^{1-\alpha}} \right) = (1-\alpha) \sum_{i=1}^{n} p_i \log_2 \frac{p_i}{q_i}$, $\alpha > 0, \alpha \neq 1$, it follows that $\sum_{i=1}^{n} p_i \log_2 \left( \frac{p_i}{q_i} \right) = 0$. Hence $p_i = q_i$.

Proposition 2.4. Let $\alpha > 0, \alpha \neq 1$ be a specified real constant and with the above professed conventions, for $n \geq 1$ an integer, the succeeding inequality is continuously accurate:

$$\sum_{i=1}^{n} x_i^{\alpha} y_i^{1-\alpha} \log_2 \left( \frac{y_i}{x_i} \right)^{\alpha-1} \leq \left( \sum_{i=1}^{n} x_i^{\alpha} y_i^{1-\alpha} \right) \log_2 \left( \frac{\sum_{i=1}^{n} x_i}{\sum_{i=1}^{n} x_i^{\alpha} y_i^{1-\alpha}} \right)$$

(2.7)

If $n = 1$, then (2.7) holds as an equality. If $n \geq 2$, then the emblem of equivalence in (2.7) holds iff $\frac{x_1}{y_1} = \frac{x_2}{y_2} = \ldots = \frac{x_n}{y_n}$.

Proof. If $n = 1$, then the insignia of equivalence clamps in (2.7) as both sides of it reduce to $(\alpha - 1) x_1^{\alpha} y_1^{1-\alpha} \log_2 \left( \frac{y_1}{x_1} \right)$. Now consider $n \geq 2$. In this case, by Theorem 2.7, we acquire

$$\left( \sum_{i=1}^{n} x_i^{\alpha} y_i^{1-\alpha} \right) \log_2 \left( \frac{\sum_{i=1}^{n} x_i^{\alpha} y_i^{1-\alpha}}{\sum_{i=1}^{n} x_i} \right) \leq \sum_{i=1}^{n} x_i^{\alpha} y_i^{1-\alpha} \log_2 \left( \frac{x_i^{\alpha} y_i^{1-\alpha}}{x_i} \right)$$

(2.8)

with the insignia of equivalence in (2.8) iff $\frac{x_1^{\alpha} y_1^{1-\alpha}}{x_1} = \frac{x_2^{\alpha} y_2^{1-\alpha}}{x_2} = \ldots = \frac{x_n^{\alpha} y_n^{1-\alpha}}{x_n}$ or equivalently, $\frac{x_1}{y_1} = \frac{x_2}{y_2} = \ldots = \frac{x_n}{y_n}$ as $\alpha > 0, \alpha \neq 1$. The inequality (2.7) follows from (2.8).

Now, we deliver arguments for the importance of (2.7) in the field of information theory.

For all $P \in \Gamma_n^*, Q \in \Gamma_n^*, n \geq 2$ a prearranged integer, we demonstrate that $D_{\alpha}(P; Q)$ is a non-decreasing function of $\alpha$, $\alpha > 0, \alpha \neq 1$. Indeed,

$$\frac{d}{d\alpha} D_{\alpha}(P; Q) = \frac{1}{(1-\alpha)^2} \left[ -\frac{\sum_{i=1}^{n} p_i^{\alpha} q_i^{1-\alpha} \log_2 \left( \frac{q_i}{p_i} \right)^{\alpha-1}}{\sum_{i=1}^{n} p_i^{\alpha} q_i^{1-\alpha}} - \log_2 \sum_{i=1}^{n} p_i^{\alpha} q_i^{1-\alpha} \right]$$

(2.9)

If in equation (2.7), we take $x_i = p_i, y_i = q_i, i = 1, 2, \ldots, n$ such that $P \in \Gamma_n^*, Q \in \Gamma_n^*$, then the term within brackets on the right hand side of (2.9) is a nonnegative real number.

Consequently, $\frac{d}{d\alpha} D_{\alpha}(P; Q) \geq 0$. Hence, $D_{\alpha}(P; Q)$ is a non-decreasing function of $\alpha$.

If there happen indices $i$ and $j$, such that $\frac{x_i}{y_i} \neq \frac{x_j}{y_j}$, then strict inequality holds in (2.7). Consequently, if $\frac{p_i}{q_i} \neq \frac{p_j}{q_j}$ for some indices $i$ and $j$, $1 \leq i \neq j \leq n$, then the term within brackets on the right hand side of (2.9) is a positive real number.

Accordingly, $\frac{d}{d\alpha} D_{\alpha}(P; Q) > 0$. Hence $D_{\alpha}(P; Q)$ is a strictly monotonic increasing function of $\alpha$ and as usual, it can certainly be ascertained that

$$D_{\alpha}(P; Q) < D(P; Q) \text{ if } 0 < \alpha < 1$$

(2.10)

and

$$D_\alpha(P; Q) > D(P; Q) \text{ if } \alpha > 1 \tag{2.11}$$

**Definition 2.5**. Through the above professed conventions and for $n \geq 1$ an integer, if the inequality $\sum_{i=1}^{n} x_i \geq \sum_{i=1}^{n} y_i$ clutches good, then the directed divergence of order $\alpha, \alpha > 0, \alpha \neq 1$, denoted by $\overline{D}_\alpha(X; Y)$ of the sequence $X$ from $Y$ is demarcated subsequently:

$$\overline{D}_\alpha(X; Y) = \frac{1}{\alpha - 1} \log_2 \left( \frac{\sum_{i=1}^{n} x_i^\alpha y_i^{1-\alpha}}{\sum_{i=1}^{n} x_i} \right) \tag{2.12}$$

Consider the case $n = 1$. In this case, by assumption, $x_1 \geq y_1$. Now from equation (2.12), we acquire the subsequent communication:
$\overline{D}_\alpha(x_1; y_1) = \log_2 \left( \frac{x_1}{y_1} \right) \geq 0$v with the insignia of equivalence iff $x_1 = y_1$.

Suppose $n \geq 2$. Making practice of the conjecture $\sum_{i=1}^{n} x_i \geq \sum_{i=1}^{n} y_i$, the equations (2.25) and (2.26), we achieve the succeeding manifestation:
$\overline{D}_\alpha(X; Y) > 0$ unless $\frac{x_1}{y_1} = ... = \frac{x_n}{y_n}$.
Now we demonstrate that $\overline{D}_\alpha(X; Y) = 0$ iff $x_i = y_i$.
If $x_i = y_i$, then $\overline{D}_\alpha(X; Y) = 0$ follows from (4.12).
Now suppose $\overline{D}_\alpha(x_1, x_2, ..., x_n; y_1, y_2, ..., y_n) = 0$. Replacing $y_i$ by $\left( \frac{x_i}{y_i} \right)^{\alpha-1}$ in (1.18) and using the fact that $\log_2 \left( \frac{x_i}{y_i} \right)^{\alpha-1} = (\alpha - 1) \log_2 \left( \frac{x_i}{y_i} \right)$, we obtain the subsequent appearance:

$$(\alpha - 1) \frac{\sum_{i=1}^{n} x_i \log_2 \left( \frac{x_i}{y_i} \right)}{\sum_{i=1}^{n} x_i} \leq \log_2 \left( \frac{\sum_{i=1}^{n} x_i^\alpha y_i^{1-\alpha}}{\sum_{i=1}^{n} x_i} \right) \tag{2.13}$$

Consider the case $\alpha > 1$. Upon employing the equations (2.5), (2.11), (2.12), (2.13) and the assumption $\sum_{i=1}^{n} x_i \geq \sum_{i=1}^{n} y_i$, we acquire the consequent exterior:

$$0 \leq \log_2 \left( \frac{\sum_{i=1}^{n} x_i}{\sum_{i=1}^{n} y_i} \right) \leq \frac{\sum_{i=1}^{n} x_i \log_2 \left( \frac{x_i}{y_i} \right)}{\sum_{i=1}^{n} x_i} \leq \overline{D}_\alpha(x_1, x_2, ..., x_n; y_1, y_2, ..., y_n) \tag{2.14}$$

Since $\overline{D}_\alpha(X; Y) = 0$, equation (2.14) contributes with the subsequent appearance:

$$\frac{\sum_{i=1}^{n} x_i \log_2 \left( \frac{x_i}{y_i} \right)}{\sum_{i=1}^{n} x_i} = 0 \text{ and } \sum_{i=1}^{n} x_i = \sum_{i=1}^{n} y_i.$$

Hence $\frac{x_1}{y_1} = \frac{x_2}{y_2} = ... = \frac{x_n}{y_n}$ and $\sum_{i=1}^{n} x_i = \sum_{i=1}^{n} y_i$. Consequently $x_i = y_i$.

The case $0 < \alpha < 1$ can be discussed similarly. Consequently, we have demonstrated that $\overline{D}_\alpha(X;Y) \geq 0$ with the insignia of impartiality iff $x_i = y_i$.

It is observed that all the measures $D(P;Q)$, $\overline{D}(X;Y)$, $D_\alpha(P;Q)$ and $\overline{D}_\alpha(X;Y)$, $\alpha > 0$, $\alpha \neq 1$ are additive.

## Applications to shannon entropy and inaccuracy measures

Let us write

$$D_1(P;Q) = I_1(P;Q) - H_1(P) \tag{2.15}$$

Motivated by this development, we may express some more measures of directed divergence Consider

$$\overline{\overline{D}}_\alpha(P;Q) = I_\alpha(P;Q) - H_\alpha(P) \tag{2.16}$$

where $\alpha > 0$, $\alpha \neq 1$. Since the right hand side of (2.16) may be a negative real number, $\overline{\overline{D}}_\alpha(P;Q)$ does not seem to be a satisfactory measure of directed divergence.

Now, consider the subsequent manifestation:

$$D^\alpha(P;Q) = I^\alpha(P;Q) - H^\alpha(P) \tag{2.17}$$

where $\alpha > 0$, $\alpha \neq 1$. Here, too, the right hand side of (2.17) may be a negative real number and consequently, $D^\alpha(P;Q)$ also does not seem to be satisfactory measure of directed divergence.

Let $d : \Gamma_n^* \times \Gamma_n^* \to R$, $n \geq 1$ a given integer be a measure of directed divergence.

Define $S : \Gamma_n^* \times \Gamma_n^* \to R$ as

$$S(P;Q) = \min(d(P;Q); d(Q;P))$$

Then $S$ is a symmetric measure of directed divergence satisfying (2.1), (2.2) and (2.3).

Now define $S^* : \Gamma_n^* \times \Gamma_n^* \to R$ as

$$S^*(P;Q) = d(P;Q) + d(Q;P) \tag{2.18}$$

Then $S^*$ is also a symmetric measure of directed divergence between the probability distributions $P \in \Gamma_n^*$, $Q \in \Gamma_n^*$. An important situation arises if we choose $d = D$ where $D : \Gamma_n^* \times \Gamma_n^* \to R$ is the Kullback-Liebler's [19] divergence defined by (2.4). In this case, writing $J$ in place of $S^*$, we have

## Symmetric divergence measures

$$J(P;Q) = \sum_{i=1}^{n} p_i \log_2\left(\frac{p_i}{q_i}\right) + \sum_{i=1}^{n} q_i \log_2\left(\frac{q_i}{p_i}\right) \tag{2.19}$$

The symmetric measure $J(P;Q)$ is usually called J-divergence [31]. It can be written as

$$J(P;Q) = \sum_{i=1}^{n} (p_i - q_i) \log_2\left(\frac{p_i}{q_i}\right) \tag{2.20}$$

Making use of (2.15), the J-divergence $J(P; Q)$ can be written as

$$J(P; Q) = [I_1(P; Q) - H_1(P)] + [I_1(Q; P) - H_1(Q)]$$
(2.21)

For the sake of notational convenience, we write $J(P; Q)$ as $J(P; Q)$ and also write (2.21) as

$$J_1(P; Q) = I_1(P; Q) + I_1(Q; P) - H_1(P) - H_1(Q)$$
(2.22)

Corresponding to (2.22), Nath [16,32] developed the subsequent manifestation:

$$J_\alpha(P; Q) = I_\alpha(P; Q) + I_\alpha(Q; P) - H_\alpha(P) - H_\alpha(Q)$$
(2.23)

where $\alpha > 0$, $\alpha \neq 1$. If $n = 1$, then $J_\alpha(1; 1) = 0$. Now consider $n \geq 2$.

It is obvious that $J_\alpha(P; Q) > 0$ unless $p_i = q_i$, $i = 1, 2,..., n$.

If $p_i = q_i$, $i = 1,..., n$; then $J_\alpha(P; Q) = 0$.

Consequently, $J_\alpha(P; Q) \geq 0$

Now suppose $J_\alpha(P; Q) = 0$. We prove that $p_i = q_i$, $i = 1, 2,..., n$. The case $n = 1$ is trivial. We restrict our discussion to the case when $n \geq 2$. In this case,

$$0 = (1-\alpha)^{-1}\log_2\left(\sum_{i=1}^{n} p_i q_i^{\alpha-1}\right) + (1-\alpha)^{-1}\log_2\left(\sum_{i=1}^{n} q_i p_i^{\alpha-1}\right) - (1-\alpha)^{-1}\log_2\left(\sum_{i=1}^{n} p_i^{\alpha}\right)$$
$$- (1-\alpha)^{-1}\log_2\left(\sum_{i=1}^{n} q_i^{\alpha}\right)$$

It stretches the subsequent communication:

$$\left(\sum_{i=1}^{n} p_i q_i^{\alpha-1}\right)\left(\sum_{i=1}^{n} q_i p_i^{\alpha-1}\right) = \left(\sum_{i=1}^{n} p_i^{\alpha}\right)\left(\sum_{i=1}^{n} q_i^{\alpha}\right)$$

Accordingly, we must have $\frac{p_1}{q_1} = \frac{p_2}{q_2} = ... = \frac{p_n}{q_n}$.

But $\sum_{i=1}^{n} p_i = 1 = \sum_{i=1}^{n} q_i$. Consequently, $p_i = q_i$. Since the right hand side of (2.23) is symmetric in $p_i$ and $q_i$; it follows that

$$J_\alpha(P; Q) = J_\alpha(Q; P)$$

Hence $J_\alpha(P; Q)$ is a symmetric measure of directed divergence between the probability distributions $P \in \Gamma_n^*$, $Q \in \Gamma_n^*$.

**Practical example: (Application to image fusion — numerical illustration)**

Consider a simple image-fusion scenario where two source images ($I_1$) and ($I_2$) produce discrete pixel-intensity histograms that are modelled by probability vectors

$$P = (p_1, \ldots, p_8) = (0.25, 0.15, 0.20, 0.10, 0.08, 0.07, 0.10, 0.05),$$

$$Q = (q_1, \ldots, q_8) = (0.20, 0.18, 0.22, 0.12, 0.06, 0.06, 0.10, 0.06).$$

Here P and Q approximate pixel intensity distributions from different sensors or bands. We compute standard divergence measures between P and Q to illustrate how the inequalities and monotonicity results in this paper apply in practice.

1. Kullback–Leibler divergence (natural logarithm, "nats"):

$$D(P\|Q) = \sum_{i=1}^{8} p_i \ln\frac{p_i}{q_i}, \quad D(Q\|P) = \sum_{i=1}^{8} q_i \ln\frac{q_i}{p_i}$$

Numerical values (computed term-by-term) give

$$D(P\|Q) \cong 0.01583250 \text{ nats} \quad D(Q\|P) \cong 0.01546532 \text{ nats,}$$

hence the symmetric J-divergence

$$J(P,Q) = D(P\|Q) + D(Q\|P) \cong 0.03129782 \text{ nats}$$

2. Rényi divergence of order $\alpha$ (we use the conventional definition)

$$D_\alpha(P;Q) = (\alpha-1)^{-1}\log_2\left(\sum_{i=1}^{n} p_i^\alpha q_i^{1-\alpha}\right); \alpha > 0, \alpha \neq 1$$

For $\alpha = 0.5$ and $\alpha = 2$ we obtain $D_{0.5}(P\|Q) \approx 0.00783223 \text{ nats}$ $\quad D_2(P\|Q) \approx 0.03212978 \text{ nats}$.

**Monotonicity check and interpretation**

For these P and Q we observe

$$D_{0.5}(P\|Q) \approx 0.00783 \leq D(P\|Q) \approx 0.01583 \leq D_2(P\|Q) \approx 0.03213,$$

which illustrates the monotonicity of the Rényi divergence in $\alpha$ (the inequalities derived in the text). In the image-fusion context a small divergence (all values are small here) indicates that the fused image will retain characteristics of the original images with relatively little information loss; thus the inequalities and bounds we derived provide a quantitative basis for comparing fusion strategies and for choosing parameters in fusion algorithms.

**Per-bin computation (illustrative terms)**

Selected per-bin terms (rounded) used in the KL and Rényi computations:

| Bin | $p_i$ | $q_i$ | $p_i * ln\left(\frac{p_i}{q_i}\right)$ | $q_i * ln\left(\frac{q_i}{p_i}\right)$ | $p_i^{0.5} * q_i^{0.5}$ | $p_i^2 * q_i^{-1}$ |
|---|---|---|---|---|---|---|
| 1 | 0.25000 | 0.20000 | 0.05578589 | −0.04462871 | 0.22360680 | 0.31250000 |
| 2 | 0.15000 | 0.18000 | −0.02734823 | 0.03281788 | 0.16431677 | 0.12500000 |
| 3 | 0.20000 | 0.22000 | −0.01906204 | 0.02096824 | 0.20976177 | 0.18181818 |
| 4 | 0.10000 | 0.12000 | −0.01823216 | 0.02187859 | 0.10954451 | 0.08333333 |
| 5 | 0.08000 | 0.06000 | 0.02301457 | −0.01726092 | 0.06928203 | 0.10666667 |
| 6 | 0.07000 | 0.06000 | 0.01079055 | −0.00924904 | 0.06480741 | 0.08166667 |
| 7 | 0.10000 | 0.10000 | 0.00000000 | 0.00000000 | 0.10000000 | 0.10000000 |
| 8 | 0.05000 | 0.06000 | −0.00911608 | 0.01093929 | 0.05477226 | 0.04166667 |

These per-bin terms sum to the KL and the internal sums used in Rényi computations; shown here to document and reproduce the numbers

To continue further, our prerequisite is the subsequent consequence:

Result 2.9. If $a$ and $b$ be positive and unequal real numbers, then the succeeding inequalities are continuously correct:

$$ra^{r-1}(a-b) > a^r - b^r > rb^{r-1}(a-b) \text{ if } r < 0 \text{ or } r > 1 \tag{2.24}$$

$$ra^{r-1}(a-b) < a^r - b^r < rb^{r-1}(a-b) \text{ if } 0 < r < 1 \tag{2.25}$$

The insignia of egalitarianism in, (2.24) and (2.25), grips when $r = 0$ or $r = 1$ or $a = b$.

Corresponding to (2.22), let us contemplate the succeeding communication:

$$J^\alpha(P;Q) = I^\alpha(P;Q) + I^\alpha(Q;P) - H^\alpha(P) - H^\alpha(Q) \tag{2.26}$$

where $\alpha > 0$, $\alpha \neq 1$.

Making practice of (1.3) and (1.5), (1.17), we acquire the succeeding manifestation:

$$J^\alpha(P;Q) = (1-2^{1-\alpha})^{-1} \left[ \sum_{i=1}^{n} p_i^\alpha + \sum_{i=1}^{n} q_i^\alpha - \sum_{i=1}^{n} p_i q_i^{\alpha-1} - \sum_{i=1}^{n} q_i p_i^{\alpha-1} \right] \tag{2.27}$$

where $\alpha > 0$, $\alpha \neq 1$. If $n = 1$, then (2.27) gives $J^\alpha(1;1) = 0$. Now suppose $n \geq 2$. Consider any index $i$, $1 \leq i \leq n$. If $p_i = q_i$, then

$(1-2^{1-\alpha})^{-1} \left[ p_i^\alpha + q_i^\alpha - p_i q_i^{\alpha-1} - q_i p_i^{\alpha-1} \right] = 0$ for all $\alpha > 0$, $\alpha \neq 1$. If $p_i \neq q_i$, $1 \leq i \leq n$, then by equations (2.24) and (2.25), we achieve the subsequent inequations:

$\alpha p_i^{\alpha-1}(p_i - q_i) > \alpha q_i^{\alpha-1}(p_i - q_i)$ if $\alpha > 1$
$\alpha p_i^{\alpha-1}(p_i - q_i) < \alpha q_i^{\alpha-1}(p_i - q_i)$ if $0 < \alpha < 1$.

Henceforth, we acquire the succeeding communications:

$p_i^\alpha + q_i^\alpha > p_i q_i^{\alpha-1} + q_i p_i^{\alpha-1}$ if $\alpha > 1$
$p_i^\alpha + q_i^\alpha < p_i q_i^{\alpha-1} + q_i p_i^{\alpha-1}$ if $0 < \alpha < 1$.

Also $(1-2^{1-\alpha}) > 0$ if $\alpha > 1$ and $(1-2^{1-\alpha}) < 0$ if $0 < \alpha < 1$.

Consequently, we acquire

$(1-2^{1-\alpha})^{-1} \left[ p_i^\alpha + q_i^\alpha - p_i q_i^{\alpha-1} - q_i p_i^{\alpha-1} \right] > 0$ for all $\alpha > 0$, $\alpha \neq 1$.

Thus, for any index $i$, $1 \leq i \leq n$, $(1-2^{1-\alpha})^{-1} \left[ p_i^\alpha + q_i^\alpha - p_i q_i^{\alpha-1} - q_i p_i^{\alpha-1} \right] \geq 0$ for all $\alpha > 0$, $\alpha \neq 1$.

Therefore $J^\alpha(P;Q) \geq 0$ this corroborates the consequence.

Now we prove that $J^\alpha(P;Q) = 0$ iff $p_i = q_i$. Indeed, from the above comprehensive conversation, it is obvious that if $p_i = q_i$; then $J^\alpha(P;Q) = 0$. Now suppose that $J^\alpha(P;Q) = 0$. We shall prove that $p_i = q_i$. The case $n = 1$ is trivial.

Now, we consider $n \geq 2$. From the above wide-spread arguments, it is clear that $J^\alpha(P;Q) = 0$, is possible when $p_i^\alpha + q_i^\alpha - p_i q_i^{\alpha-1} - q_i p_i^{\alpha-1} = 0$, that is, $p_i^{\alpha-1}(p_i - q_i) = q_i^{\alpha-1}(p_i - q_i)$. Making use of Result 2.9, this is possible only when $p_i = q_i$, as $\alpha \neq 0$ and $\alpha \neq 1$. Since the right hand side of (2.27) is symmetric in $p_i$ and $q_i$; it follows that

$$J^\alpha(P;Q) = J^\alpha(Q;P)$$

Consequently, for all $\alpha > 0$, $\alpha \neq 1$, $J^\alpha(P;Q)$ is a symmetric measure of directed divergence.

## Conclusion

Inequalities in information theory are supposed to be superfluous pillars in the realm of communication and data dispensation. Their prominence lies not only in their mathematical sophistication but likewise in their real-world solicitations across miscellaneous disciplines. As we traverse a period of speedy technological progression and accumulative dependence on interconnected classifications, inequalities in information theory make available a compass for manipulative communication infrastructures that can withstand the complexities of the contemporary world. By addressing the encounters pretended by noise, restricted bandwidth, and security apprehensions; these inequalities pave the approach for tough and proficient information conversation, eventually influencing the forthcoming landscape of communication machineries. The inequalities, theorems, definitions, and divergence-related communications presented in this study are useful in the field of information theory. All inequalities and related communications those are successful for positive real discovered by commissioning more discrete divergence models.

## Author contributions

**Conceptualization:** Vikramjeet Singh, Sunil Kumar Sharma, Mona Bin-Asfour.

**Funding acquisition:** Sunil Kumar Sharma.

**Methodology:** Vikramjeet Singh, Sunil Kumar Sharma, Om Parkash, Mona Bin-Asfour.

**Resources:** Vikramjeet Singh.

**Supervision:** Vikramjeet Singh, Om Parkash.

**Validation:** Vikramjeet Singh, Sunil Kumar Sharma.

**Writing – original draft:** Vikramjeet Singh, Sunil Kumar Sharma, Mona Bin-Asfour.

**Writing – review & editing:** Vikramjeet Singh, Om Parkash.

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
