## [Decision Letter · Decision Letter 0]

7 Oct 2025

Numerous Inequalities and related Communications accompanying Discrete Divergence Models in Probability Spaces

PLOS ONE

Dear Dr. Singh,

Thank you for submitting your manuscript to PLOS ONE. After careful consideration, we feel that it has merit but does not fully meet PLOS ONE’s publication criteria as it currently stands. Therefore, we invite you to submit a revised version of the manuscript that addresses the points raised during the review process.

We look forward to receiving your revised manuscript.

Kind regards,

Fucai Lin, Ph.D.

Academic Editor

PLOS ONE

Journal Requirements:

“The author extends the appreciation to the Deanship of Postgraduate Studies and Scientific Research at Majmaah University for funding this research work through the project number (ICR-2024-1260).”

Please state what role the funders took in the study.  If the funders had no role, please state: 'The funders had no role in study design, data collection and analysis, decision to publish, or preparation of the manuscript.'

“The authors have declared that no competing interests exist.

I have read the journal's policy and the authors of this manuscript have the following competing interests:”

Please confirm that this does not alter your adherence to all PLOS ONE policies on sharing data and materials, by including the following statement: 'This does not alter our adherence to  PLOS ONE policies on sharing data and materials.” (as detailed online in our guide for authors http://journals.plos.org/plosone/s/competing-interests). If there are restrictions on sharing of data and/or materials, please state these. Please note that we cannot proceed with consideration of your article until this information has been declared.

5. Please provide a complete Data Availability Statement in the submission form, ensuring you include all necessary access information or a reason for why you are unable to make your data freely accessible. If your research concerns only data provided within your submission, please write 'All data are in the manuscript and/or supporting information files' as your Data Availability Statement.

7. Thank you for stating the following in the Acknowledgments Section of your manuscript:

“The author extends the appreciation to the Deanship of Postgraduate Studies and Scientific Research at Majmaah University for funding this research work through the project number (ICR-2024-1260).”

“The author extends the appreciation to the Deanship of Postgraduate Studies and Scientific Research at Majmaah University for funding this research work through the project number (ICR-2024-1260).”

Reviewer's Responses to Questions

**Comments to the Author**

1. Is the manuscript technically sound, and do the data support the conclusions?

Reviewer #1: Yes

Reviewer #2: No

2. Has the statistical analysis been performed appropriately and rigorously?

Reviewer #1: Yes

Reviewer #2: N/A

3. Have the authors made all data underlying the findings in their manuscript fully available?

Reviewer #1: Yes

Reviewer #2: Yes

4. Is the manuscript presented in an intelligible fashion and written in standard English?

Reviewer #1: No

Reviewer #2: No

Reviewer #1: For the manuscript titled:

“Numerous Inequalities and Related Communications Accompanying Discrete Divergence Models in Probability Spaces”

Review Report Recommendation: Accept with Minor Revisions 1. Overall Assessment

This manuscript presents a mathematically rigorous and conceptually rich investigation into a wide array of inequalities connected with discrete divergence models in probability spaces. The authors make a substantive contribution to information theory by expanding the theoretical underpinnings of divergence measures such as the Kullback-Leibler divergence, Rényi divergence, and other generalized models. The presentation is dense yet meticulous, reflecting a deep engagement with foundational and contemporary literature in information theory, statistical divergence, and entropy-based modeling.

2. Strengths

Original Contribution:

The paper offers a comprehensive and novel treatment of inequalities that have practical significance in quantifying divergence and inaccuracy in discrete probability distributions. Notably, the extension and synthesis of divergence models enhance the interpretability and applicability in domains such as coding theory, statistical inference, and data compression.

Mathematical Rigor:

The manuscript is grounded in solid mathematical formalism, providing well-structured theorems, definitions, and proofs. The logical progression from entropy formulations to inequalities and divergence implications is coherent and substantiated.

Interdisciplinary Relevance:

The discussion highlights applications of entropy-based models in diverse areas such as image processing, machine learning, statistical modeling, and medical signal analysis. This interdisciplinary scope broadens the potential impact of the work.

Clarity of Objectives:

The aim of the paper—to establish new inequalities and connect them with known divergence measures—is clearly defined and systematically addressed throughout the paper.

3. Suggestions for Improvement

While the paper is commendable, the following minor revisions are suggested for improved clarity and broader accessibility:

Language and Style:

Certain sentences in the abstract and introduction are stylistically dense and could benefit from grammatical refinement and simplification. For example, the phrase "the analysis of inequalities allows for the recording of ideal coding schemes" could be rephrased to "the analysis of inequalities aids in the formulation of optimal coding schemes."

Notation and Formatting:

Some mathematical expressions would benefit from clearer formatting, particularly in longer equations. The use of consistent notation for divergence functions and entropy should be ensured throughout.

Contextual Examples:

The inclusion of one or two applied examples—perhaps in the domain of image fusion, data classification, or sensor networks—would greatly help to illustrate the real-world utility of the proposed inequalities and divergence models.

Section Titles and Flow:

Breaking down the longer sections into more focused subsections (e.g., "Applications to Shannon Entropy," "Generalized Divergence Functions," "Symmetric Divergence Measures") could enhance readability.

4. Technical Accuracy

The theoretical results appear to be correct, and the proofs are logically sound. The use of auxiliary inequalities, such as Hölder-type or logarithmic convexity properties, is appropriate and well-grounded. References to classical results (Shannon, Rényi, Kullback-Leibler) and modern elaborations (e.g., f-divergence measures, J-divergence) are apt and well-integrated into the discourse.

5. Conclusion

This is a highly valuable and well-executed manuscript that deserves publication in PLOS ONE following minor linguistic and structural revisions. The authors have not only deepened our understanding of divergence in probability spaces but also provided tools with implications for theoretical and applied research across multiple domains.

Reviewer #2: While the manuscript offers a comprehensive mathematical treatment of inequalities related to discrete divergence models, the scope is very theoretical and dense. The introduction and discussion could improve clarity by adding more intuitive explanations and examples of practical applications to help readers from applied fields grasp the significance.

The work builds upon known divergence measures such as Kullback-Leibler and Renyi divergences. However, the manuscript would benefit from more explicit highlighting of its novel contributions, distinguishing its new inequalities or generalizations clearly from prior art.

Some sections are overly technical, with long dense formulas and theoretical proofs. The manuscript would gain from better structuring, including summarizing key results in tables or diagrams, and separating theoretical developments from applied implications more distinctly.

The referencing is extensive but could be more critically integrated. It is suggested to better contextualize the new inequalities within existing frameworks and to discuss more recent related works, if any, to position the study in the current research landscape.

The paper is heavily theoretical without experimental validation, simulations, or examples showing the practical utility of the new inequalities in real-world scenarios. Including such elements would substantially strengthen the impact.

Although mathematically rigorous, some parts of the manuscript suffer from verbose and complicated language, which could be simplified for readability without losing precision

**Do you want your identity to be public for this peer review?** For information about this choice, including consent withdrawal, please see our Privacy Policy

Reviewer #1: No

Reviewer #2: No

---

## [Decision Letter · Decision Letter 1]

6 Jan 2026

Numerous Inequalities and related Communications accompanying Discrete Divergence Models in Probability Spaces

PLOS One

Dear Dr. Singh,

Thank you for submitting your manuscript to PLOS ONE. After careful consideration, we feel that it has merit but does not fully meet PLOS ONE’s publication criteria as it currently stands. Therefore, we invite you to submit a revised version of the manuscript that addresses the points raised during the review process.

We look forward to receiving your revised manuscript.

Kind regards,

Fucai Lin, Ph.D.

Academic Editor

PLOS One

Journal Requirements:

Reviewer's Responses to Questions

**Comments to the Author**

Reviewer #2: All comments have been addressed

Reviewer #3: All comments have been addressed

2. Is the manuscript technically sound, and do the data support the conclusions?

Reviewer #2: Yes

Reviewer #3: Yes

3. Has the statistical analysis been performed appropriately and rigorously?

Reviewer #2: (No Response)

Reviewer #3: Yes

4. Have the authors made all data underlying the findings in their manuscript fully available?

Reviewer #2: Yes

Reviewer #3: Yes

5. Is the manuscript presented in an intelligible fashion and written in standard English?

Reviewer #2: Yes

Reviewer #3: Yes

Reviewer #2: All comments have been appropriately addressed, and the necessary revisions have been thoroughly implemented. Therefore, I recommend that the manuscript be accepted for publication.

Reviewer #3: Summary and Overall:

Well written abstract, introduction, background, presentation of results and conclusion.

Major comments:

The aim of the study is clearly stated in the abstract. The paper focuses on the analysis of inequalities that are used in information theory. It extends known inequalities in information theory and introduces a general divergence model in general probability spaces. The paper shows a smooth transition towards the aim of the study.

The authors supported the paper theoretical outcomes through the application to Shannon Entropy and inaccuracy measures. As well as through numerical illustration through applying the results to image fusion.

Language and Typos:

The language of the paper sounds good now. However, I have noticed two typos:

• The first line from the paragraph after (1.5): “different instigators …”. I believe authors mean investigators.

• The last line from the last paragraph after (1.12): “To satisfy our aim, we need the following standard findings from Ash[1] and Cover and Thomas [2]:”. The reader expects to read about the findings on the next page, however; a new section is presented on that page.

• “Result 2.9: If a and b be positive..”. Correct “be” to “are”.

**Do you want your identity to be public for this peer review?** For information about this choice, including consent withdrawal, please see our Privacy Policy

Reviewer #2: No

Reviewer #3: No

---

## [Editor Report · Decision Letter 2]

12 Jan 2026

Numerous Inequalities and related Communications accompanying Discrete Divergence Models in Probability Spaces

PONE-D-25-27194R2

Dear Dr. Singh,

We’re pleased to inform you that your manuscript has been judged scientifically suitable for publication and will be formally accepted for publication once it meets all outstanding technical requirements.

Kind regards,

Fucai Lin, Ph.D.

Academic Editor

PLOS One
---

## [Editor Report · Acceptance letter]

PONE-D-25-27194R2

PLOS One

Dear Dr. Singh,

I'm pleased to inform you that your manuscript has been deemed suitable for publication in PLOS One. Congratulations! Your manuscript is now being handed over to our production team.

Kind regards,

on behalf of

Professor Fucai Lin

Academic Editor

PLOS One